# Empirical vs. light-use efficiency modelling for estimating carbon fluxes in a mid-succession ecosystem developed on abandoned karst grassland

**Koffi Dodji Noumonvi** *, Mitja Ferlan

Department of Forest Ecology, Slovenian Forestry Institute, Ljubljana, Slovenia

* noumonvikoffidodji@hotmail.fr

## Abstract

Karst systems represent an important carbon sink worldwide. However, several phenomena such as the $CO_2$ degassing and the exchange of cave air return a considerable amount of $CO_2$ to the atmosphere. It is therefore of paramount importance to understand the contribution of the ecosystem to the carbon budget of karst areas. In this study conducted in a mid-succession ecosystem developed on abandoned karst grassland, two types of model were assessed, estimating the gross primary production (GPP) or the net ecosystem exchange (NEE) based on seven years of eddy covariance data (2013–2019): (1) a quadratic vegetation index-based empirical model with five alternative vegetation indices as proxies of GPP and NEE, and (2) the vegetation photosynthesis model (VPM) which is a light use efficiency model to estimate only GPP. The Enhanced Vegetation Index (EVI) was the best proxy for NEE whereas SAVI performed very similarly to EVI in the case of GPP in the empirical model setting. The empirical model performed better than the VPM model which tended to underestimate GPP. Therefore, for this ecosystem, we suggest the use of the empirical model provided that the quadratic relationship observed persists. However, the VPM model would be a good alternative under a changing climate, as it is rooted in the understanding of the photosynthesis process, if the scalars it involves could be improved to better estimate GPP.

## Introduction

Within the context of global changes, the exchanges of carbon between the atmosphere and ecosystems are of high importance for a better understanding of the biosphere's carbon balance. An accurate quantification of carbon fluxes is essential for carbon budget studies at large scales [1]. When grasslands used as pasture are abandoned, they naturally evolve into woody systems and ultimately into forests [2], which leads to a change in the ecosystem carbon balance. This change is more beneficial in karst ecosystems where the $CO_2$ originating from the degassing of caves is absorbed more efficiently by plants, hence improving the carbon budget of such ecosystems [3].

**Data Availability Statement:** All the data are publicly accessible at https://doi.org/10.6084/m9.figshare.12520778.v1.

**Funding:** This research was funded by the European Research Council under the European Union's Horizon 2020 research and innovation program under grant agreement No. 774234, and also by the Slovenian Research Agency, projects Z4-8217 and J4-9297 and research program P4-0107.

**Competing interests:** The authors have declared that no competing interests exist.

The carbon balance of the ecosystem expressed as Net Ecosystem Exchange (NEE), being the difference between the gross primary production (GPP) and the ecosystem respiration ($R_{eco}$), is often measured by the eddy covariance (EC) method. However, due to the maintenance costs of an EC tower and its restriction to a limited footprint, large scale carbon fluxes estimations must rely on remote sensing information. Vegetation indices (VI) have a good empirical relationship with the fraction of absorbed photosynthetically active radiation (fAPAR) [4–6]. Hence, they are often used as proxy of GPP.

For the estimation of GPP, remote sensing can be used through the application of different models [7,8], such as VI-based empirical models [9–11] and light-use efficiency (LUE)-based models [5,12–14]. VI-based empirical models rely mostly on regression whereas LUE models are rooted in the LUE theory which states that GPP is related to the absorbed photosynthetically active radiation (APAR) and the LUE or photosynthetic efficiency by unit of photosynthetically active radiation (PAR). LUE is obtained by reducing a maximum photosynthetic efficiency ($LUE_{max}$) under environmental stress factors such as low/high temperatures or water stress [4,5].

While several studies applied LUE models to estimate GPP, the estimation of NEE is usually more critical since it also includes $R_{eco}$ [15]. In fact, $R_{eco}$ is one of the main sources of uncertainty in the estimation of NEE [16]. To overcome this limitation, some studies attempted to estimate NEE and GPP directly with simple empirical regression models especially in grasslands and croplands [10,11]. For evergreen systems where the seasonal decrease in photosynthetic activity can occur without substantial declines in canopy greenness reflected by VI, this approach is inadequate [17]. What about intermediary ecosystems with tree patches mixed with a grassland? Would a VI-based empirical model help estimate NEE and GPP or do we need to rely only on other approaches such as a LUE model?

The present study applied an empirical modelling approach to assess NEE in a mid-succession ecosystem developed on abandoned karst grassland. By applying the same for GPP, the empirical approach was compared to a LUE model, the vegetation photosynthesis model (VPM). The specific objectives were:

1. to compare the performance of different VI derived from Landsat images for estimating NEE and GPP using the empirical model,

2. to compare the performance of the empirical model and the LUE model in assessing GPP.

## Materials and methods

### Study site

This study concerned a mid-succession ecosystem developed on abandoned karst grassland in the Podgorski Kras plateau situated in the south-western part of Slovenia (longitude 13° 55' 0.12" E; latitude 45° 32' 36.564" N). Despite a history marked by anthropic actions such as intensive agriculture and grazing which led to eroded landscapes, the near abandonment of agricultural practices that resulted from the economic development favoured the development of different stages of vegetation succession, from grasslands to secondary oak forests. The soil, made essentially of insoluble fractions of carbonates that originated from the karst phenomena on a limestone bedrock, is superficial [18]. The climate is sub-Mediterranean, with minimum and maximum mean daily temperatures of 1.8°C and 19.9°C respectively in January and June. The mean annual temperature is 10.5°C, and the average annual precipitation is about 1370 mm. These are statistics of 30 years (1971–2000) climate data from four different meteorological stations of the area [11,19]. The study area (Fig 1) covers a diverse ecosystem where both

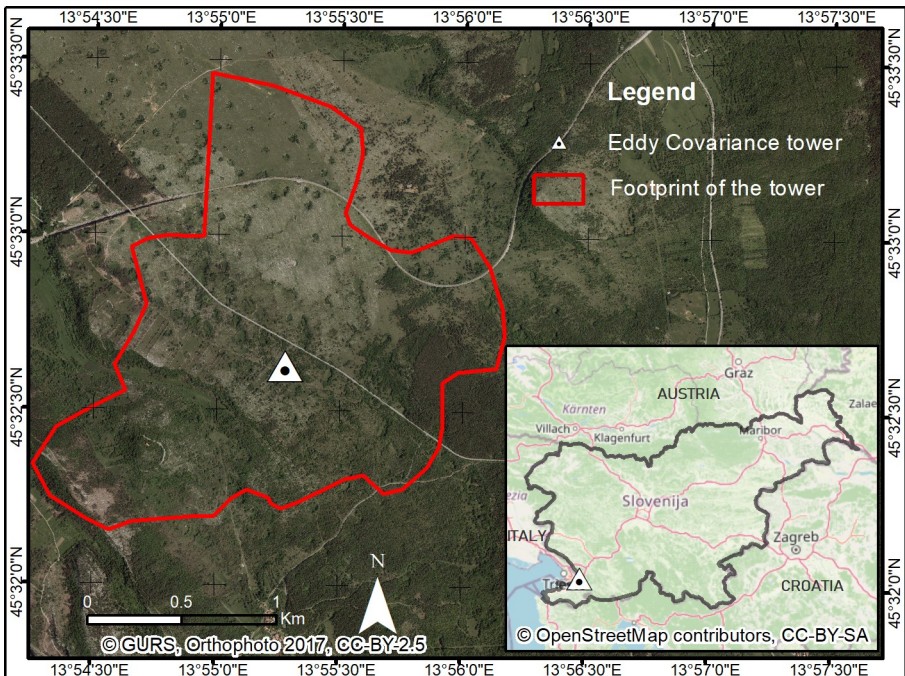

**Fig 1. Study area.** In red is the footprint of the EC tower (location marked with triangle) estimated in a previous study [18,20].

grass, shrubs and tree patches coexist. Although pure grasslands still cover a non-negligible part, in the last 30 years, over 20% of the former grasslands were transformed into mid and late forest succession, with a mixture of grass, shrubs and trees. Woody plants encompass shrubs of early successional stage and tree species, mainly *Quercus pubescens* Willd [18]. In a previous study [18], the EC tower footprint (Fig 1) was estimated based on the contribution of fluxes from a given distance from the tower [20], and the mean distance from where the tower monitors 90% of fluxes was estimated to 1530 m. In the estimated footprint, grass/shrubs and trees can be roughly estimated to cover 70% and 30% respectively, considering also grass under tree patches.

## Data acquisition

**Eddy covariance and meteorological data.** An EC system installed in 2008 at the study site (Fig 1) provides continuous measurements of $CO_2$ and $H_2O$ exchanges between the ecosystem and the atmosphere. More information about instrumentation can be found in a previous study [21]. Post processing of raw EC data was done in the EddyPro v6.2.1. software. The weather station installed along with the EC system measures environmental variables including soil temperature and soil water content (SWC) at 10 cm depth, incident global radiation (Rg), air temperature (Tair), air humidity and precipitation (P). A gap-filling has been performed for Tair and Rg based on nearby weather stations in Koper and Skocjan (located at 15 and 14 km from the tower, respectively). NEE data were gap-filled and partitioned into GPP and $R_{eco}$ as in a previous study [22]. The gap-filling and partitioning were done using the REddyProc online tool [23]. Additionally to the regular weather station measurements at the EC tower, a setting (LI-190, Li-Cor, Lincoln, NE USA) installed above the ecosystem, below the trees and above grassland allowed to measure respectively the incoming photosynthetic photon flux density (incPPFD, two sensors), the transmitted or below tree canopy PPFD (bcPPFD,

two sensors) and the non-absorbed PPFD by the grass (naPPFD, one sensor), that can be used to establish the relationship between fAPAR and VI [4–6].

**Spectral vegetation indices.** In this study, Landsat 8 operational land imager (OLI) images (30m resolution) between April 2013 and December 2019 were processed to compute five VI (Table 1) using google earth engine. Since we used surface reflectance data made available by the United States Geological Survey [24], the only processing carried out consisted in masking out clouds, cloud shadow and snow before computing the VI with the formulas written in Table 1. The downloaded VI clipped to the extent of the EC tower footprint were then used to create half-monthly composite VI (1st to 15th of the month and 16th to end of the month). Composite VI images with more than 5% of masked pixels in the EC tower footprint were discarded before averaging the VI.

## NEE and GPP estimation models

**Empirical model.** Multiple model comparisons suggested a quadratic model better fits the relationship observed between GPP or NEE as response variable and VI as explanatory variable in this study. The empirical model applied to estimate GPP and NEE is therefore a quadratic model:

$$GPP \text{ or } NEE = a*VI^2 + b*VI + c \qquad (1)$$

Where VI represents the five alternative vegetation indices; a, b and c are regression constants.

**Vegetation photosynthesis model.** The second model applied is the VPM model which is a LUE model [30]. Three modifiers were considered: a water scalar, a temperature scalar and a phenological scalar. The VPM model as applied in this study can be written as follows:

$$GPP = PAR*fAPAR*LUE_{max}*W_{scalar}*T_{scalar}*P_{scalar} \qquad (2)$$

Where GPP is the gross primary production (in $gC.m^{-2}.halfmonth^{-1}$), PAR is the photosynthetically active radiation (in $MJ.m^{-2}.halfmonth^{-1}$), assumed here to be a constant fraction (45%) of Rg [31,32]; fAPAR is the fraction of absorbed photosynthetically active radiation by the ecosystem; $LUE_{max}$ (in $gC.MJ^{-1}$) is the potential LUE for the investigated ecosystem under ideal environmental conditions; $W_{scalar}$, $P_{scalar}$ and $T_{scalar}$ represent modifiers for water, phenology and temperature, respectively. They account for the reduction of photosynthetic activity under water and temperature stress according to the phenological stage of the vegetation. The scalars range from 0 (non-vegetation phenophase, water or temperature is a limiting

**Table 1. Vegetation indices explored in this study.**

| Index | Formula | Reference |
|:---:|:---:|:---:|
| **NDVI** | (NIR–R) / (NIR + R) | [25] |
| **GNDVI** | (NIR—G) / (NIR + G) | [26] |
| **EVI** | (2.5 * (NIR–R)) / (NIR + 6*R– 7.5*B + 1) | [27] |
| **SAVI** | ((1 + L) (NIR–R)) / (NIR + R + L) | [28] |
| **LSWI** | (NIR–SWIR1) / (NIR + SWIR1) | [29] |

L is a vegetation cover-dependent constant ranging from 0 (i.e. very green vegetation) to 1 (i.e. areas with no green vegetation), considered = 0.5 here. B, G, R, NIR and SWIR1 represent respectively the blue (0.452–0.512 μm), green (0.533–0.590 μm), red (0.636–0.673 μm), near infrared (0.851–0.879 μm), and shortwave infrared (1.566–1.651 μm) bands of the Landsat 8 image.

factor for photosynthesis) to 1 (phenological stage, water or temperature conditions are ideal for photosynthesis).

**Estimation of fAPAR from EVI.** PPFD measurements available for the year 2019 were used to compute fAPAR as in a previous study [33].

$$fAPAR = 1 - \frac{PPFD_{out}}{incPPFD} \tag{3}$$

Where incPPFD is the incoming PPFD and PPFD_out is the average below canopy PPFD for trees (bcPPFD) or the non-absorbed PPFD for grass (naPPFD).

fAPAR was computed separately for grass and trees, and then averaged with weights of 70% and 30%, respectively, to account for their relative landcover in the tower footprint. Afterwards, fAPAR was aggregated half-monthly like EVI, and a linear regression was performed to establish the relationship between fAPAR and EVI in 2019. Finally, the established relationship was applied to estimate fAPAR from EVI for the entire timeframe of the study (2013–2019).

*Estimation of LUE_{max}.* LUE_{max} was determined as the slope of the linear regression through origin between GPP elaborated from EC and APAR for midday fluxes (11 am to 1 pm) of uncloudy days during the growing season, as in a previous study [34]:

$$LUE_{max} = \frac{\overline{GPP}}{\overline{APAR}}, \tag{4}$$

Where $\overline{GPP}$ is the gross primary production (in $gC.m^{-2}.30min^{-1}$) and $\overline{APAR}$ is the absorbed photosynthetically active radiation (in $MJ.m^{-2}.30min^{-1}$) obtained by multiplying fAPAR by PAR.

The dataset used for the estimation of LUE_{max} was filtered to only use midday half-hourly GPP values from the growing season (April to October) and when SWC is higher than 0.146 m3.m-3 and the VPD lower than 20 hPa, which are the non-drought conditions in our ecosystem as reported in a recent study [35]. Uncloudy days were identified by finding days where the curve of PPFD per half-hour from 7 am to 3 pm has a near-perfect downward quadratic shape ($y = x+x^2$, $r^2 > 0.95$).

*Estimation of the modifiers.* The modifiers were estimated as in previous studies [36,37]. The water scalar was estimated as follows:

$$W_{scalar} = \frac{1 + LSWI}{1 + LSWImax}, \tag{5}$$

Where LSWI is the Land Surface Water Index and LSWI_{max} is the maximum value of LSWI over the entire seven years, during the growing seasons.

The phenological scalar ($P_{scalar}$) was also estimated from LSWI as developed in a previous study [12]. Unlike in its original definition [36] where the phenological scalar was set to 1 during the wet phase due to the retention of leaves during that period, $P_{scalar}$ was left as calculated in this study, due to the heterogeneity of the vegetation.

$$P_{scalar} = \frac{1 + LSWI}{2}, \tag{6}$$

The temperature scalar was estimated as described in a previous study [38]:

$$T_{scalar} = \frac{(T - Tmin)(T - Tmax)}{[(T - Tmin)(T - Tmax) - (T - Topt)^2]}, \tag{7}$$

Where $T_{opt}$, $T_{min}$, and $T_{max}$ are optimal, minimum, and maximum air temperatures (˚C) for photosynthesis, respectively and T is the average air temperature (from 11 am to 4 pm). $T_{opt}$ was set to 20˚C, $T_{min}$ and $T_{max}$ were set to 0 and 35˚C respectively, and $T_{scalar}$ was set to 0 when air temperature goes beyond these limits [39,40].

## Data analysis

Gap-filled NEE and partitioned GPP as well as Rg were summed-up to obtain the half-monthly aggregates, temporally matching the VI aggregates. For air temperature however, we used the average to obtain half-monthly aggregates. Only temperatures measured between 11 am—4 pm were considered during the aggregation, because of a lot of noise caused by temperature scalar when whole day air temperature was considered. For the empirical model development, we used the data splitting approach, which consisted in splitting the dataset randomly into a training set (70%) and a validation set (30%). This was repeated a thousand times, and average regression coefficients were obtained as well as average regression accuracy metrics such as the coefficient of determination ($r^2$), the root mean square error (rmse) and the p-value, both from training (Train) and validation (Test) sets, with their confidence intervals. Additionally, the corrected Akaike Information Criterion (AICc) was computed for each empirical regression model using the "AICcmodavg" package in R. The choice of AICc over AIC was motivated by the fact that AICc is corrected for small samples, and converges towards AIC for large samples, making it always suitable for model selection. AICc differences (ΔAICc) were computed between each candidate model and the model with the lowest AICc to support the other accuracy metrics in choosing the best model. A model can be said to outperform the others if the ΔAICc with all the other models is higher than 2, i.e. if the presumed best model has a AICc lower than that of all the other models with at least 2 units of difference [41]. For the VPM model, all the parameters were computed, and GPP was estimated. In order to correct the estimated GPP that was found to be underestimated differently for the growing season (April to October) and the non-growing season (November to March), correction factors were computed in the same multiple data splitting setting described previously for the empirical model, on the same training and validation sets. The growing and non-growing seasons were treated differently and for each growing phase, the training dataset was used to estimate the correction factors, and the validation dataset was used to assess the performance of the model through accuracy metrics calculation. The comparison of the best empirical model for GPP estimation with the VPM model was based on ΔAICc as described for the empirical model selection. All the data were analysed with the R software v3.5.1.

## Results

### Carbon fluxes from the eddy covariance tower and satellite-derived vegetation indices

Carbon fluxes from the EC tower and VI used in this study are reported in Fig 2. High photosynthetic activities occur in the studied ecosystem between May and July and is translated into large negative values of NEE (Fig 2A), large values of GPP and Reco (Fig 2B) and generally high values of VI (Fig 2C). In contrast, low photosynthetic activity could be observed from October to March where GPP values are low and $R_{eco}$ reflects significantly in NEE values that become mostly positive. A gap due to equipment failure and too large to be filled could be noticed in fluxes during the year 2018 from May to August. All the VI considered in this study show similar trends that match well with the trend of GPP. They all reach their maximum

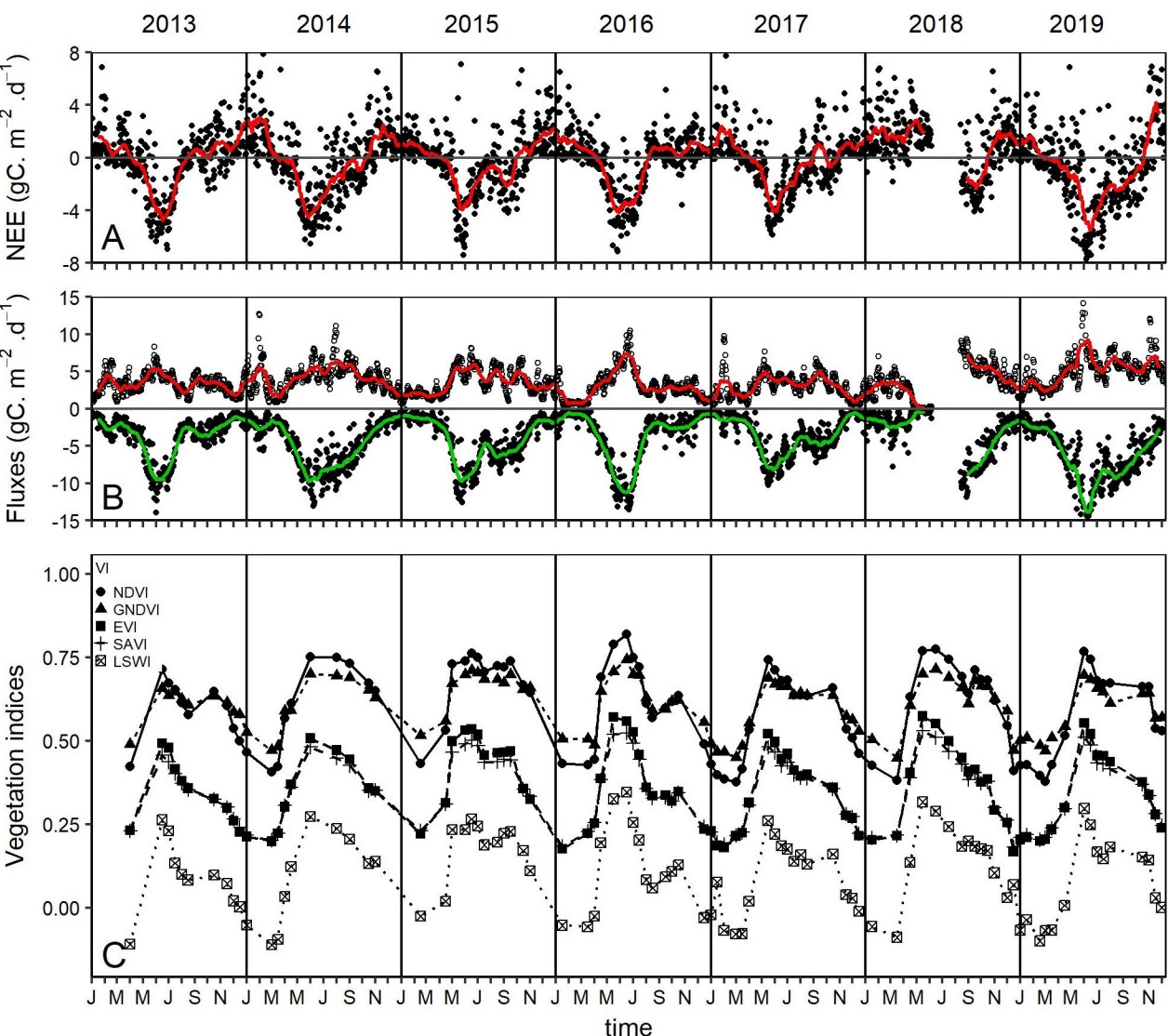

**Fig 2. Fluxes & VI used in the study.** (A) daily aggregates of NEE, (B) daily aggregates of partitioned GPP (negative values) and R$_{eco}$ (positive values), and (C) Vegetation indices (NDVI, GNDVI, EVI, SAVI and LSWI). The red and green lines are the rolling means of the corresponding fluxes.

about the same time as GPP (between May and July) and reach their minimum almost at the end of the non-growing season (around February).

## Empirical model performance in estimating GPP and NEE

Results obtained from fitting the quadratic model to measured NEE and partitioned GPP are presented in Fig 3. More information about the average "Train" and "Test" accuracy metrics (rmse, r$^2$, p-value and AICc) were obtained from the multiple data splitting setting (Table 2). It appears that EVI is the best proxy for both NEE and GPP in this ecosystem with r$^2$ of 0.73 and 0.82, respectively. In the case of GPP, SAVI showed performances comparable to that of EVI ($\Delta$AICc < 2), whereas other VI such as NDVI and GNDVI performed less. NEE generally showed a lower correlation with all VI, compared to GPP.

The "Train" and "Test" accuracy metrics are comparable, which leads to conclude that there was neither an over-fitting nor under-fitting of the model. In addition, all correlations

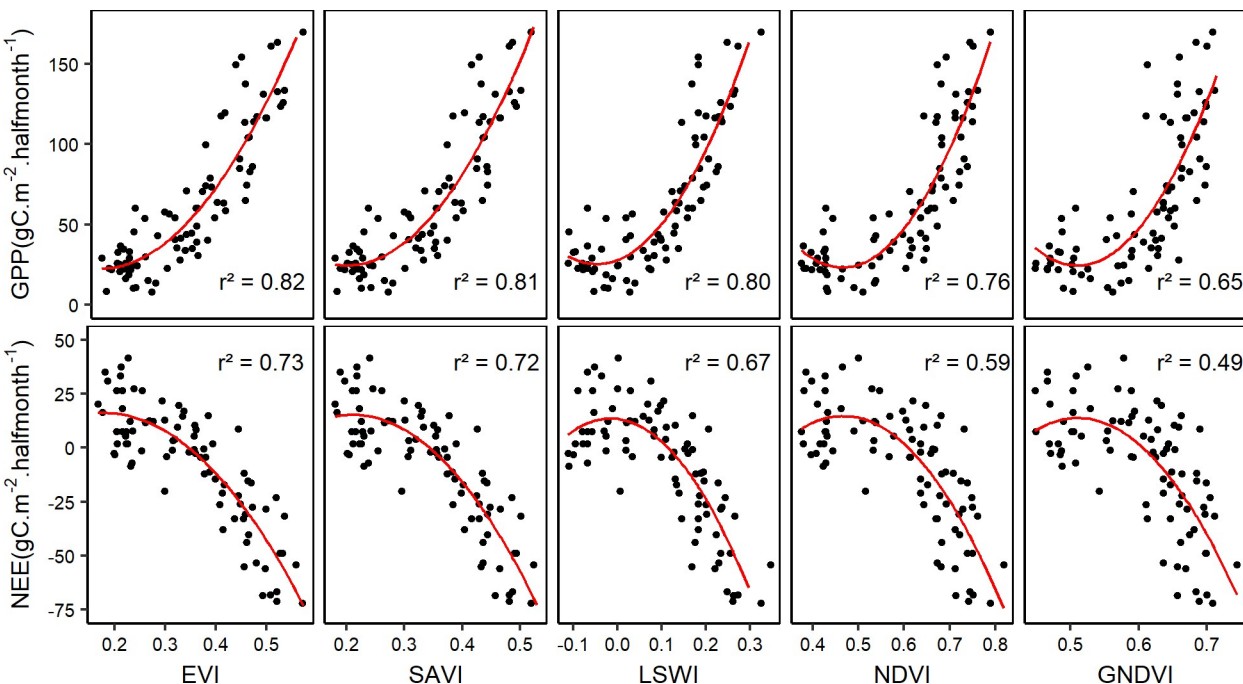

**Fig 3. Empirical model fitting of fluxes with VI.** The red lines represent quadratic regression lines fitted to the cloud of points (flux~VI).

were very significant (both "Train" and "Test" p-values < 0.001) with the only exception of NEE with GNDVI, where the correlation is still significant (p-value "Test" < 0.01) despite the relatively low $r^2$ value.

## VPM model performance in estimating GPP

**Estimated fAPAR.** Based on the PPFD records in 2019, the calculated fAPAR shows that there is an important difference between grass and tree phenology (Fig 4A). The grass species

**Table 2. Accuracy metrics of the empirical model.**

| Flux | VI | Equation | rmse_test | $r^2$_test | p_test | rmse_train | $r^2$_train | p_train | AICc | ΔAICc |
|------|-----|----------|-----------|------------|--------|------------|-------------|---------|------|-------|
| **GPP** | **EVI** | **919.41 * EVI$^2$–298.13 * EVI + 46.58** | **19.08±5.31** | **0.82±0.12** | **\*\*\*** | **18.58±2.11** | **0.83±0.04** | **\*\*\*** | **541.02±50.73** | **0.00** |
| **GPP** | **SAVI** | **1425.73 * SAVI$^2$–571.3 * SAVI + 81.66** | **19.32±5.38** | **0.81±0.12** | **\*\*\*** | **18.84±2.13** | **0.82±0.04** | **\*\*\*** | **542.74±50.85** | **1.72** |
| GPP | LSWI | 1177.37 * LSWI$^2$ + 108.36 * LSWI + 27.81 | 20.09±6.81 | 0.8±0.14 | *** | 19.67±2.73 | 0.81±0.05 | *** | 547.91±52.21 | 6.89 |
| GPP | NDVI | 1335.65 * NDVI$^2$–1241.75 * NDVI + 311.81 | 21.89±6.46 | 0.76±0.15 | *** | 21.56±2.55 | 0.77±0.06 | *** | 559.25±52.69 | 18.23 |
| GPP | GNDVI | 2873.12 * GNDVI$^2$–2934.78 * GNDVI + 773.92 | 26.76±8.5 | 0.65±0.22 | *** | 26.25±3.4 | 0.65±0.09 | *** | 583.36±55.19 | 42.35 |
| **NEE** | **EVI** | **574.74 * EVI$^2$–206.4 * EVI + 2.47** | **14.54±3.1** | **0.73±0.15** | **\*\*\*** | **13.85±1.16** | **0.75±0.05** | **\*\*\*** | **498.92±47.39** | **0.00** |
| NEE | SAVI | 874.28 * SAVI$^2$–370.08 * SAVI + 23.97 | 14.96±3.26 | 0.72±0.15 | *** | 14.24±1.22 | 0.73±0.05 | *** | 502.25±47.77 | 3.33 |
| NEE | LSWI | 804.61 * LSWI$^2$ + 24.89 * LSWI—13.33 | 16.59±4.35 | 0.67±0.19 | *** | 15.63±1.56 | 0.68±0.07 | *** | 513.52±48.91 | 14.60 |
| NEE | NDVI | 718.89 * NDVI$^2$–670.56 * NDVI + 141.62 | 18.18±4.21 | 0.59±0.21 | *** | 17.44±1.57 | 0.6±0.07 | *** | 526.76±50.04 | 27.84 |
| NEE | GNDVI | 1516.99 * GNDVI$^2$–1550.72 * GNDVI + 382.49 | 20.18±4.53 | 0.49±0.22 | ** | 19.41±1.69 | 0.5±0.08 | *** | 539.79±51.1 | 40.87 |

The suffixes "_test" and "_train" refer respectively to the accuracy metrics computed on the test/validation dataset and the training dataset, with their confidence interval. The best selection of GPP and NEE regressions are highlighted in bold. p_values were transformed into correlation significance, i.e.

*** means a high significance (p_value < 0.001), and

** means a medium significance (p_value < 0.01).

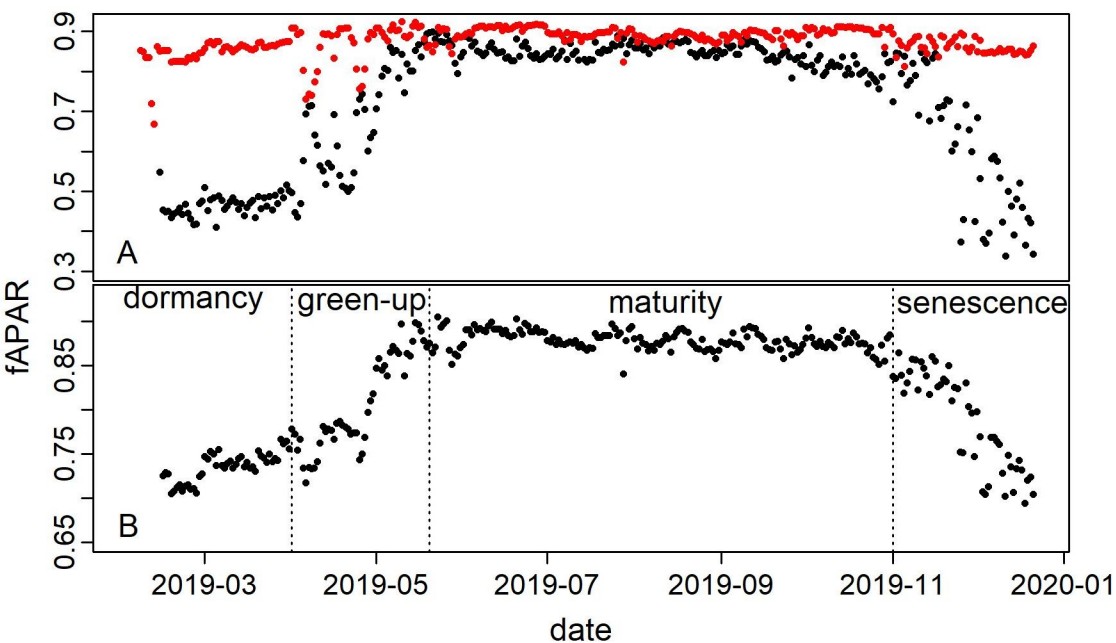

**Fig 4. Daily fraction of absorbed PAR (fAPAR) in 2019.** (A) daily grass (red) and tree (black) fAPAR. (B) daily ecosystem average fAPAR, assuming 70% and 30% cover of grass and trees, respectively. The phenophases (dormancy, green-up, maturity and senescence) refer to development stages of the deciduous tree species of the study area.

did not show a clear change in PAR absorption throughout the year, whereas fAPAR for tree patches showed a typical trend that can be used to split the considered period into four phases. Data was only available from February, but a dormancy can be noticed until March, characterized by a relatively low and steady absorption of PAR. In our ecosystem in 2019, the vegetation started to absorb more PAR in April till the second half of May, which can be called the green-up phase. This phase is characterized by an important increase in fAPAR. The vegetation reaches maturity towards end of May after which the fAPAR stays almost constant. Finally, early in November, fAPAR starts to decrease more substantially through winter, and this phase can be called senescence. These phenological phases, although mostly related to the tree species, are reflected in the average ecosystem fAPAR (Fig 4B).

The relationship between NDVI or EVI and fAPAR is generally considered to be linear or near-linear [34]. The established quadratic relationship between fAPAR and EVI in this study (Fig 5), with a high $r^2$ (0.93) and a relatively low rmse (0.09), made it possible to estimate fAPAR for years with no available PPFD measurements for direct fAPAR calculation.

## Estimated LUE$_{max}$

The concept of LUE, which is the amount of carbon produced by APAR unit is usually assumed to have a maximum value (LUE$_{max}$), representing the maximum conversion rate of APAR into biomass under optimal environmental conditions (i.e. Tair, SWC, VPD and PAR). The relationship between midday GPP and APAR for the growing season during uncloudy days (Fig 6) allowed to estimate LUE$_{max}$ to 0.85 gC.MJ$^{-1}$.

**GPP estimates.** The VPM model was applied on the entire dataset after preparing all the required inputs (fAPAR, PAR, LUE$_{max}$, W$_{scalar}$, T$_{scalar}$ and P$_{scalar}$) described earlier. The model output shows a clear difference between growing and non-growing season (Fig 7A). Additionally, there is an underestimation of GPP by the VPM model differently according to the

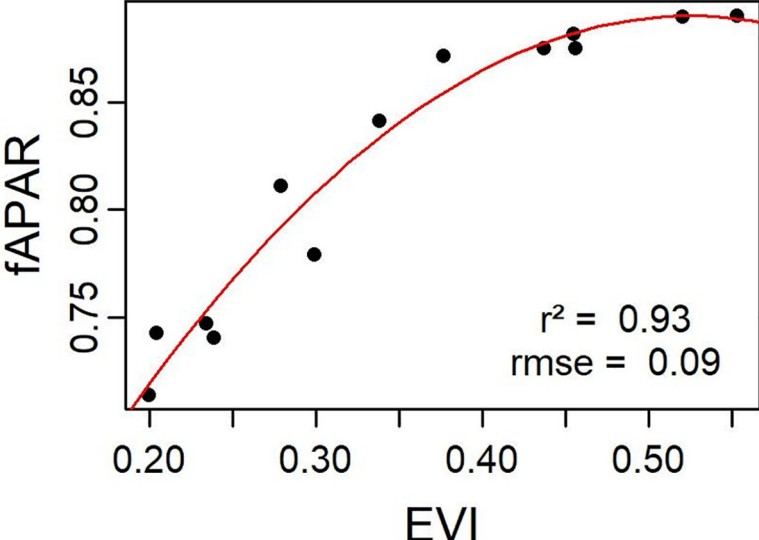

**Fig 5. Relationship between half-monthly fAPAR and EVI.** The red line is the quadratic regression line fitted to the cloud of points. The resulting equation is: fAPAR = -1.62 * $EVI^2$ + 1.70 * EVI + 0.44.

growing phase of the vegetation. Therefore, a seasonal correction factor was determined through multiple data splitting, and the final VPM model was validated by computing "Train" and "Test" accuracy metrics (Table 3). The wider confidence interval obtained for the correction factor during the non-growing season (November to March) reflects a larger fluctuation partly due to the low amount of data available during that period.

The average correction factors obtained from the training sets are 2 ± 0.08 and 2.51 ± 0.37 respectively for the growing and non-growing seasons. The corrected estimated GPP values were plotted against observed values (Fig 7B), with an overall coefficient of determination of 0.77.

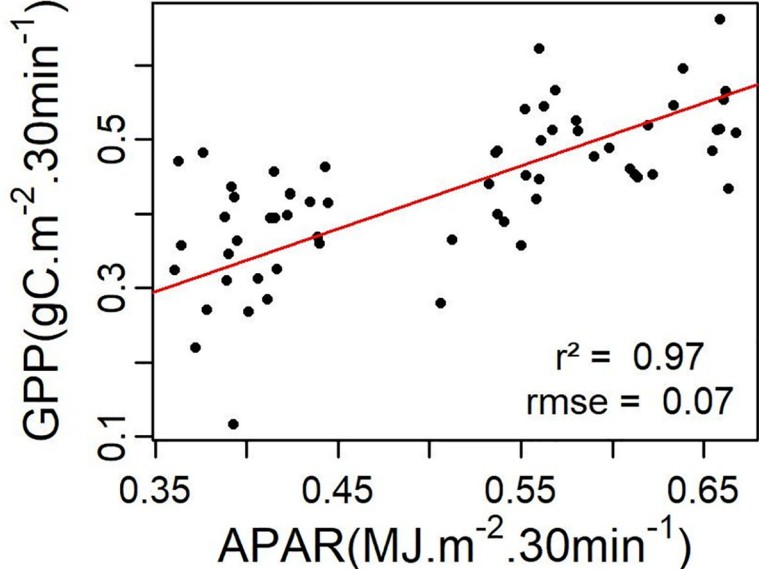

**Fig 6. Relationship between half-hourly GPP and APAR.** The red line is the linear regression line through origin fitted to the cloud of points (black dots), representing midday data for uncloudy days during the growing season. The resulting equation is: GPP = 0.85 * APAR.

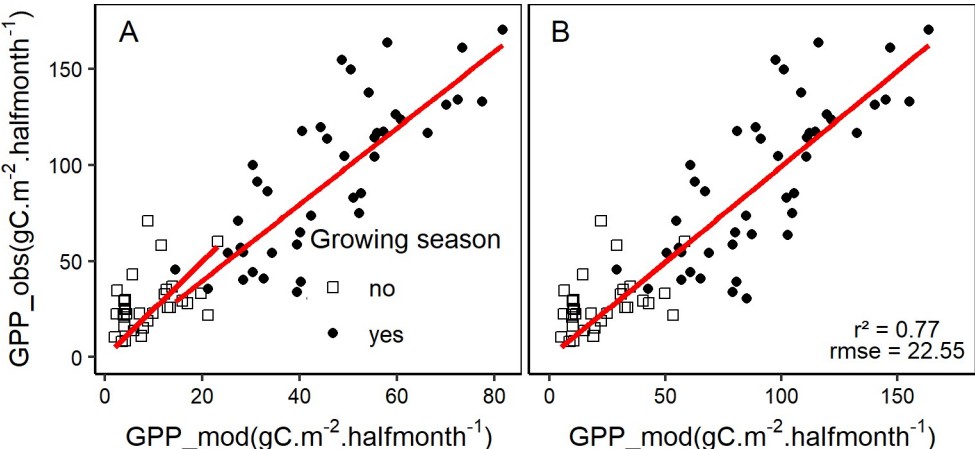

**Fig 7. Observed vs. modelled GPP using the VPM model.** (A) uncorrected VPM output, (B) seasonally corrected VPM output (correction factors of 2 and 2.51 for growing and non-growing season, respectively).

## Best models selection

In the light of Table 2, the best empirical model for NEE was the one with EVI as proxy since the ΔAICc was higher than 2 between the best model (with EVI as proxy) and the other models. In the case of GPP however, there was no substantial difference between the model with the lowest AICc (with EVI as proxy of GPP) and the model based on SAVI since the ΔAICc between these two models was lower than 2. SAVI can therefore be used as an alternative to EVI in the estimation of GPP using the empirical model. The lower $r^2$ and higher rmse and AIC (ΔAICc > 2, Table 3) obtained in the case of the VPM model in comparison with the best empirical model for GPP suggests that the empirical model performed better than the VPM model. Based on the selection of the best models among the empirical models (Table 2) and the VPM model, estimated fluxes were plotted along with measured values (Fig 8).

## Discussion

### Empirical relationship between carbon fluxes and vegetation indices

The empirical relationship between GPP and VI has been proven in several studies, but limited mainly to grasslands and croplands [10]. Different VI proved to be good proxies for GPP in different ecosystems [42,43]. While NDVI is a widely used VI in the estimation of GPP due to its capability to discriminate seasonal changes related to photosynthetic activity [10], several other VI are being used recently, in order to find the one that better suits a particular ecosystem. Among the five VI explored for estimating GPP and NEE in the present study, EVI was found to be the best VI proxy for both GPP and NEE. This seems adequate since the ecosystem addressed is not a pure grassland as it was the case in a previous study where NDVI was the best proxy for GPP [11]. The heterogeneous nature of the ecosystem in this study, made of a

**Table 3. Accuracy metrics and seasonal correction factors for GPP estimated with the VPM model.**

| rmse_test | $r^2$_test | p_test | rmse_train | $r^2$_train | p_train | fact_non_growing | fact_growing | AICc | ΔAICc |
|---|---|---|---|---|---|---|---|---|---|
| 22.55 ± 5.91 | 0.77 ± 0.14 | *** | 22.04 ± 2.31 | 0.78 ± 0.05 | *** | 2.51 ± 0.37 | 2 ± 0.08 | 554.71 ± 74.22 | 13.69 |

The suffixes "_test" and "_train" refer respectively to the accuracy metrics computed on the test/validation and the training datasets. p_values were transformed into correlation significance, i.e. *** means a high significance (p_value < 0.001).

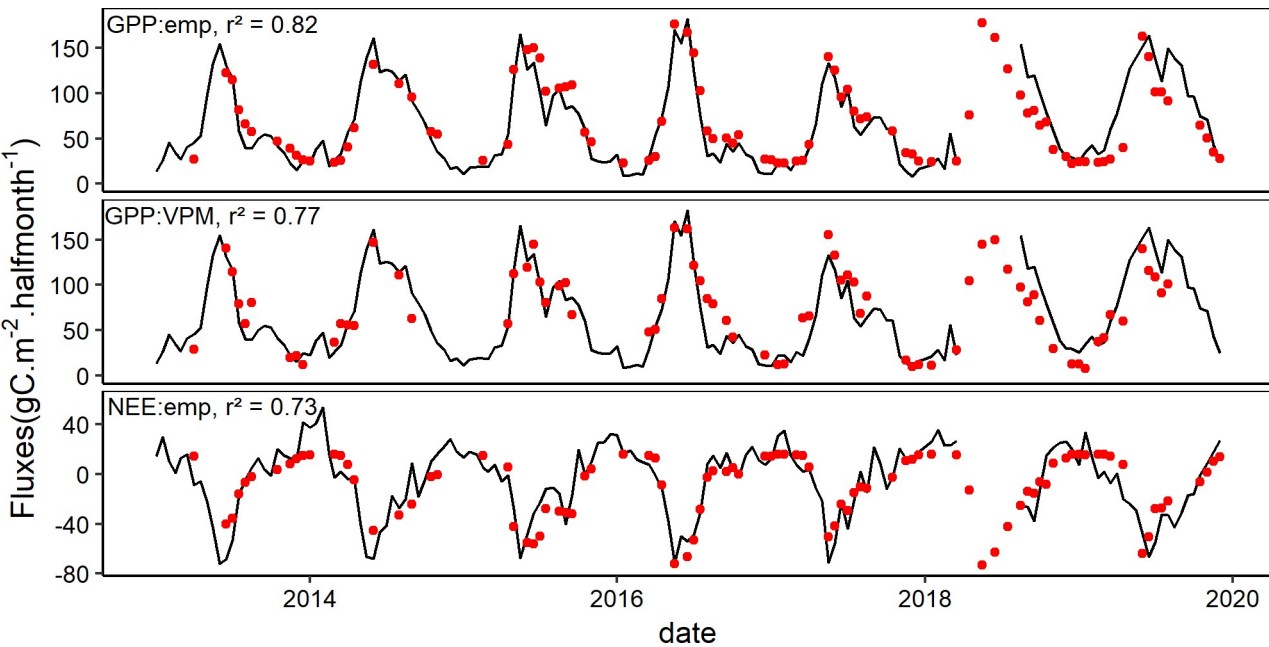

**Fig 8. Observed (black lines) and modelled (red dots) half-monthly fluxes.** The top panel represents GPP from empirical model, the middle panel represents GPP from VPM model and the bottom panel represents NEE from empirical model.

mix of tree patches and grassland, partly explains why EVI performed better than NDVI. In fact, EVI proved to be a better proxy for GPP in forest ecosystems where NDVI can get saturated [42,44]. Although $r^2$ and rmse suggested EVI to be the best proxy for GPP, the low ΔAICc between the EVI and SAVI-based models led to conclude that there was no substantial difference of performance between the two models [41]. The better performance of SAVI compared to NDVI can be supported by the fact that in some open ecosystems, the sensitivity of NDVI to soil background brightness makes it less performant [42,45], and SAVI is an optimized VI to account for soil background reflectance [28].

Several studies have proved that there exists, to a certain extent, a coupling between GPP and Reco, especially in ecosystems of shrublands [46], heathlands [47], and different forest ecosystems [48,49]. This coupling is characterized by different slopes of the GPP ~ Reco relationship, depending on vegetation species and the timescale considered, i.e. daily, monthly, or yearly. Additionally, the relationship is weak in nutrient-rich forests and very strong in nutrient-poor forests. In fact, poor soils result into lower heterotrophic respiration, which is the complex part of ecosystem respiration [50]. Our study area, being a karstic ecosystem with shallow soils and therefore relatively nutrient-poor, it was possible to also estimate half-monthly aggregates of NEE using a direct VI-based model, with EVI as the best proxy.

## fAPAR, LUE<sub>max</sub> and modifiers in the VPM model

The calculation of fAPAR requires continuous measurements of incoming and below canopy PAR, which are most of the time unavailable. In this study, we used one-year measurements of incoming and below tree canopy PPFD as well as the non-absorbed portion of PPFD by grassland, to establish a relationship with satellite-derived EVI data. Despite the low number of available pairs of "EVI, fAPAR" for only one year, the high correlation observed ($r^2$ = 0.93) confirms the previous findings on the linear relationship between fAPAR and VI. In a comparative study, EVI was found to be the best proxy for fAPAR in drought conditions in a maize

field, with $r^2 = 0.69$ [51]. A good quadratic relationship ($r^2 = 0.88$) was found between fAPAR and EVI for maize and soya fields, even though NDVI performed better in those ecosystems [52]. Thanks to the good relationship between fAPAR and EVI, the latter has been used in LUE models as a proxy of fAPAR, assuming a linear relationship across various ecosystems, from grasslands to different forest types [53].

The $LUE_{max}$ term is very important in the VPM model, as it is the starting point of LUE calculation. Based on the fact that LUE is at its maximum when plants are in their best environmental condition, considered in this study as periods during the growing season when SWC and VPD are not limiting, i.e. SWC > 0.146 m3.m-3 and VPD < 20 hPa [35], the high correlation ($r^2 = 0.97$) obtained between midday GPP and fAPAR confirms the adequacy of the estimation of $LUE_{max}$ from EC data [34,54]. $LUE_{max}$ values of 3.68 ± 1.98 gC.MJ$^{-1}$ and 0.84 ± 0.82 gC.MJ$^{-1}$ have been reported in Canadian grasslands and forests respectively [54], whereas a $LUE_{max}$ value of 1.61 gC.MJ$^{-1}$ has been reported in an alpine grassland [55]. Although reported $LUE_{max}$ values by different studies usually show large differences, the estimated $LUE_{max}$ of 0.85 gC.MJ$^{-1}$ in the ecosystem of the present study (tree patches mixed with shrubs and grass) seems to be a fair estimate. This $LUE_{max}$ value is however slightly higher than that of the biome properties look-up table for MOD17 which reports 0.768 gC.MJ$^{-1}$ for wooded grasslands and 0.8 gC.MJ$^{-1}$ for grassy woodlands [56], which can be considered as ecosystems similar to the one of our study area.

$LUE_{max}$ estimation is determinant in the accuracy of GPP estimates. The fact that the VPM model underestimated GPP and required a correction factor can be partly due to an underestimation in $LUE_{max}$. A possible underestimation of MOD17 GPP values due to an underestimation of $LUE_{max}$ has been reported in a previous study [55]. A constant $LUE_{max}$ is usually a source of error in some LUE models [57], and should rather be considered variable since it depends not only on temperature, water stress and phenology as in the VPM model, but also on plant physiological characteristics (e.g., leaf nitrogen), light intensity (e.g., diffuse radiation), and landscape feature (e.g., spatial scales) [58,59]. Another important source of error in LUE models originates from the estimation of modifiers (i.e. temperature and water scalars for instance), which can bias the resulting actual LUE. For instance, $T_{scalar}$ can be overcorrected (low values) for months with high PAR values, but low air temperatures [12], hence emphasizing the importance of a careful choice of the minimum temperature value used in the VPM model.

## Other challenges of the VPM model

The VPM model like many other LUE models has been reported to make GPP predictions that agree well with observations in some forest ecosystems such as evergreen needleleaf and deciduous broadleaf forests [5,12]. However, in other ecosystem types such as evergreen broadleaf forests and shrublands, generally low performances were observed in predicting GPP using LUE models [5]. In our ecosystem which encompasses grass and shrub species as well as deciduous forest species, the VPM model underestimated GPP differently between growing and non-growing season. The higher correction factor required for the non-growing season (end of autumn to winter) reflects a higher underestimation of GPP. While the underestimation during non-growing season can be partly explained by an overcorrection of $T_{scalar}$, errors introduced by the estimation of fAPAR [5,60] in addition to the possibility that in the studied ecosystem, LUE scalars considered in the VPM model might have overestimated the actual environmental stress-induced reduction of $LUE_{max}$, are possible reasons for the underestimation of GPP. Additionally, the heterogeneity of the study area (coexistence of grass, shrubs and tree patches) would lead to different and complex physiological processes, that

could explain the differences observed in growing vs. non-growing season VPM estimates of GPP [35,61]. This heterogeneity was observed in our study through the estimation of fAPAR, which changed very little throughout the year for grassland, whereas important changes could be noticed for tree patches. The resulting estimated average ecosystem fAPAR could also be an important source of error in the modelled GPP values. It is also important to mention flux partitioning as a possible source of error in some ecosystems [62], which can cause a discrepancy between partitioned and modelled GPP [12].

The VPM model was applied in this mixed system composed of tree patches, grass and shrubs. In this study, $P_{scalar}$ was not used exactly as in the original model in which $P_{scalar}$ is set to 1 upon full leaf expansion. The reason for that is simply the fact that we obtained better results in our ecosystem by adopting a variable $P_{scalar}$ even during the growing season, which would reflect the changing phenology and physiology of plants throughout the growing season. This is further justified by the heterogeneity of the study area, shown by the different activity patterns reflected by an almost constant fAPAR in grassland, and a seasonal fAPAR for tree patches.

## VPM vs empirical model

The best empirical model (quadratic model) obtained with EVI ($r^2$ = 0.82, rmse = 19.08 and AICc = 541.02) performed better than the VPM model after correcting GPP estimates ($r^2$ = 0.77, rmse = 22.55 and AICc = 554.71). In addition, the quadratic relationship established between NEE and EVI makes it much easier to estimate the carbon balance, without necessarily having to estimate $R_{eco}$, which is needed in the case of the VPM model. Several studies reported a better performance of VPM model over VI-based empirical models [37,63] in ecosystems such as grasslands and deciduous forests. To our knowledge, no previous studies compared the VPM and VI-based model in an ecosystem very similar to the one of this study in which vegetation (grass, shrub and tree patches) combines with the karstic heterogeneous environment to make a complex system with specific water availability conditions. In our study, the empirical (VI-based model) performed slightly better than the VPM model as the latter was found to underestimate GPP. However, under changing climate, the VPM model could capture better the uncertainty that could arise from the use of an empirical model which could fail to produce reliable GPP estimates in future if the relationship between GPP and EVI would change. A possible improvement of the VPM model in the particular case of our study area could focus on improving the estimation of the scalars by integrating the simultaneous effects of SWC, Tair and VPD on photosynthesis, in order to capture the currently unexplained error in the GPP estimates.

## Limitations of the study

In this study, VI have been used in both VPM and empirical models, in attempt to estimate GPP and NEE. To overcome the limitation of the number of available pairs of half-monthly aggregated fluxes and VI, the multiple data splitting approach has been applied, allowing to develop the models on the training set and validate the results on the validation or test set by computing different accuracy metrics. By applying a simple regression to estimate NEE, the possible complexity of the Reco component is being simplified. Therefore, more uncertainties could arise when applying the developed model to estimate NEE if the coupling of GPP and Reco will become weak in the future, which could happen if the heterotrophic part of ecosystem respiration will become more important for instance due to global warming [64]. Another limitation was related to the fact that $LUE_{max}$ was developed only based on one year (2019)

available data. However, the relatively good performance of the best models covers up for the previously mentioned limitations.

## Conclusion

Among the several existing models used to estimate carbon fluxes, a quadratic VI-based model and a LUE model, the VPM model were evaluated in a mid-succession ecosystem developed on abandoned karst grassland. It was possible to estimate both GPP and NEE with the quadratic model, whereas the VPM model underestimated GPP. The results suggest therefore the use of EVI to estimate carbon fluxes using the quadratic VI-based model. However, empirical models must be checked overtime, to make sure that the relationship persists, i.e. the regression constants have not changed. On the other hand, an improved VPM model with better estimates of LUE scalars for the area investigated in this study would be more advisable for long term application in the context of climate changes.

## Acknowledgments

The authors thank Gregor Skoberne, Klemen Eler and Polona Hafner for periodic field data collection. The authors also thank the two anonymous reviewers whose suggestions helped improve and clarify this manuscript.

## Author Contributions

**Data curation:** Mitja Ferlan.

**Formal analysis:** Koffi Dodji Noumonvi.

**Methodology:** Koffi Dodji Noumonvi.

**Writing – original draft:** Koffi Dodji Noumonvi.

**Writing – review & editing:** Koffi Dodji Noumonvi, Mitja Ferlan.

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
