## [Decision Letter · Decision Letter 0]

29 May 2020

PONE-D-20-11727

Empirical vs. light-use efficiency modelling for estimating carbon fluxes in a mid-succession ecosystem developed on abandoned karst grassland

PLOS ONE

Dear Koffi Dodji Noumonvi,

Thank you for submitting your manuscript to PLOS ONE. After careful consideration, we feel that it has merit but does not fully meet PLOS ONE’s publication criteria as it currently stands. Therefore, we invite you to submit a revised version of the manuscript that addresses the points raised during the review process.

The detailed comments from the reviewers are attached below. You will see that they have proposed some feedback to how to improve the paper including the key methods.

We look forward to receiving your revised manuscript.

Kind regards,

Wang Li

Academic Editor

PLOS ONE

3. We note that Figure 1 in your submission contains map and satellite images which may be copyrighted. All PLOS content is published under the Creative Commons Attribution License (CC BY 4.0), which means that the manuscript, images, and Supporting Information files will be freely available online, and any third party is permitted to access, download, copy, distribute, and use these materials in any way, even commercially, with proper attribution. For these reasons, we cannot publish previously copyrighted maps or satellite images created using proprietary data, such as Google software (Google Maps, Street View, and Earth). For more information, see our copyright guidelines: http://journals.plos.org/plosone/s/licenses-and-copyright.

Reviewers' comments:

Reviewer's Responses to Questions

**Comments to the Author**

1. Is the manuscript technically sound, and do the data support the conclusions?

Reviewer #1: Yes

Reviewer #2: Partly

2. Has the statistical analysis been performed appropriately and rigorously? 

Reviewer #1: No

Reviewer #2: Yes

3. Have the authors made all data underlying the findings in their manuscript fully available?

Reviewer #1: Yes

Reviewer #2: No

4. Is the manuscript presented in an intelligible fashion and written in standard English?

Reviewer #1: Yes

Reviewer #2: Yes

5. Review Comments to the Author

Reviewer #1: This study estimated GPP and NEE based on VI empirical models and compared the estimation accuracy of GPP between empirical models and VPM model in a mid-succession ecosystem developed on abandoned karst grassland. Similar attempts have been increasingly made in recent years. An interesting finding is that enhanced vegetation index was the best proxy for NEE. This study is well designed and the details of this study are well described. However, I still have several comments that need to be clarified.

1. It’s not clear to me why the authors used “in-sample” and “out-of-sample”? They are the training dataset and validation dataset, respectively?

2. Why did not you use independent data to validate the accuracy of VPM model? Although VPM GPP can be estimated directly by using several parameters, I think the validation is necessary.

3. Line 257, the subsection “3.3. Best models selection” only has one sentence, this absolutely needs to be improved and more descriptions are needed.

4. In discussion section, adding subsection will improve the readability.

5. Many studies have demonstrated the feasibility of using VI, especially the EVI, and VPM model to estimate GPP. What are the differences in your research? What are the limitations of your research methods? Why EVI can be used to estimate NEE directly in your study? I think the above issues need to be further analyzed in the discussion.

6. Lines, 280, 289, 299, 300, 305, please check whether the citation format of the reference is correct.

Reviewer #2: General comments:

The study compared the model performance of simple remote-based gross primary production (GPP) models and vegetation production model (VPM) with tower-based GPP estimation from EC observations in an abandoned karst grassland. However, some flaws need further improvement. At least, current manuscript is rough both in innovation and expression of results.

Major points:

1. Abstract: The first two sentences are off the point you want to study, because this study is a case study. Thus, maybe a bit better start is a simple and clear introduction for the importance and particularity of your research site.

2. Introduction (Lines 47-55): the same problem with the point 1, process-based models and machine learning models are outside of your research. Thus, focus the introduction on the VI-based models and LUE models for GPP estimations.

3. Results (Lines 257-259): Obviously, results of the best models selection are not showed in this part. Add some useful information in this part. In addition, considering a better model selection method because a simple comparison of statistical index (R2 and RMSE) is easy to make errors. For example, actually, tt's hard to know which model is better for two models with similar R2, just like 0.82 and 0.8, or 0.73 and 0.72. (Fig. 3)

Minor points:

1. Methods: add some information for your decision of footprint for EC-observations.

2. Lines 95-97: “could be found in [24]”, it is a wrong way for references marking. You can rewrite it as “could be found in a previous study [24]”. So many similar references errors throughout the text, please check and revise.

6. PLOS authors have the option to publish the peer review history of their article (what does this mean?). If published, this will include your full peer review and any attached files.

Reviewer #1: No

Reviewer #2: No

---

## [Author Response · Author response to Decision Letter 0]

26 Jun 2020

We would like to thank the editor and the two reviewers for their valuable comments, that certainly improved the quality of our paper. Please find below a point by point response to all issues raised during the review process. We successfully addressed each of the comments in the manuscript, and changes can be easily noticed in the revised manuscript labelled “Revised Manuscript with Track Changes”.

Editor’s comments regarding the journal’s requirements

Author response: We have carefully checked the requirements of PLOS ONE, and made the changes to meet these requirements. For instance, we have now removed the zip code and added the department name on the title page, changed the naming of figure panels in the figure legend, table legends have been moved below tables, and all borders were added to tables.

Regarding file naming, we have now checked, and in our revised submission, we named all files according to the requirements. Figure names were also checked and meet the requirements of the journal. Figures resubmitted were also checked using the PACE engine.

Heading numbers were also removed as seen in the journal’s main body format requirement.

Author response: Thank you for this reminder. The data used in the paper has now been published on Figshare, and is publicly available at the following DOI: https://doi.org/10.6084/m9.figshare.12520778.v1 .

3. We note that Figure 1 in your submission contains map and satellite images which may be copyrighted. All PLOS content is published under the Creative Commons Attribution License (CC BY 4.0), which means that the manuscript, images, and Supporting Information files will be freely available online, and any third party is permitted to access, download, copy, distribute, and use these materials in any way, even commercially, with proper attribution.

Author response: We are very sorry that we did not write the source of the basemaps used, and we thank you very much for pointing that out. We have now put the correct reference of the basemaps used in the updated Figure 1.

Originally, one of the base maps was from National geographic, and we replaced it now with Open street maps (https://www.openstreetmap.org/copyright). The other basemap is a digital orthophoto produced by the Surveying and Mapping Authority of the Republic of Slovenia (GURS) (https://www.e-prostor.gov.si/access-to-geodetic-data/ordering-data/#tab2-1791). Both basemaps are distributed under creative commons with unrestricted re-use, either non-commercially or commercially, as you can see in the links between brackets.

Therefore, I have now added the licensing type on the revised Figure 1.

Comments of the Reviewer 1

This study estimated GPP and NEE based on VI empirical models and compared the estimation accuracy of GPP between empirical models and VPM model in a mid-succession ecosystem developed on abandoned karst grassland. Similar attempts have been increasingly made in recent years. An interesting finding is that enhanced vegetation index was the best proxy for NEE. This study is well designed, and the details of this study are well described. However, I still have several comments that need to be clarified.

1. It’s not clear to me why the authors used “in-sample” and “out-of-sample”? They are the training dataset and validation dataset, respectively?

Author response: We thank the reviewer for pointing out this inconsistency in our writing. We actually meant the same thing by “in-sample” and “training dataset” for instance. We have now corrected this throughout the manuscript, for consistency, and used only “train” and “test” instead of “in-sample” and “out-of-sample”.

2. Why did not you use independent data to validate the accuracy of VPM model? Although VPM GPP can be estimated directly by using several parameters, I think the validation is necessary.

Author response: We thank the reviewer for this valuable comment. If we did not have to apply correction factors, there would be no need to split the dataset because all the observed values would have been independent dataset used to validate the model through the calculation of accuracy metrics. However, the fact that we needed to correct the estimated values with a correction factor determined using regression makes the reviewer’s suggestion very useful.

Therefore, we have now used the multiple data splitting for finalizing the VPM model as used in our study (obtaining the correction factor using the training the dataset), and validating the model with the test dataset by the same occasion. The reason why we used the multiple data splitting is to overcome the limitation of the small dataset size.

3. Line 257, the subsection “3.3. Best models selection” only has one sentence, this absolutely needs to be improved and more descriptions are needed.

Author response: This is a very useful comment, and we are sorry for having overlooked this section. We have now added some more description, regarding why the best models were said to be best, and we also compared the best empirical model with the VPM model. 

Thanks for highlighting this problem. 

4. In discussion section, adding subsection will improve the readability.

Author response: We thank the reviewer for this comment which will certainly improve the readability of our paper. We have now split the discussion into 5 sections accordingly, which are: 4.1. Empirical relationship between carbon fluxes and VI ; 4.2. fAPAR, LUEmax and modifiers in the VPM model ; 4.3. Other challenges of the VPM model ; 4.4. VPM vs Empirical model ; 4.5. Limitations of the study.

5. Many studies have demonstrated the feasibility of using VI, especially the EVI, and VPM model to estimate GPP. What are the differences in your research? What are the limitations of your research methods? Why EVI can be used to estimate NEE directly in your study? I think the above issues need to be further analyzed in the discussion.

Author response: We have now addressed these different issues in the discussion. Additionally, some of these issues which were already addressed before are more visible now after splitting the discussion into sections. The changes can be seen throughout the discussion section. We have now added a separate section that addresses the limitations of the study (section 4.5).

We thank the reviewer for this important comment.

6. Lines, 280, 289, 299, 300, 305, please check whether the citation format of the reference is correct.

Author response: Thank you very much for pointing out this issue that was also raised by the Reviewer 2. We have now reviewed the citation style so that it is scientifically correct. Wherever possible, we changed the sentence into passive voice and added the reference at the end, and when this was not possible, we added the author’s name before the shortened citation at the beginning of the sentence.

Comments of the Reviewer 2

The study compared the model performance of simple remote-based gross primary production (GPP) models and vegetation production model (VPM) with tower-based GPP estimation from EC observations in an abandoned karst grassland. However, some flaws need further improvement. At least, current manuscript is rough both in innovation and expression of results.

Major points:

1. Abstract: The first two sentences are off the point you want to study, because this study is a case study. Thus, maybe a bit better start is a simple and clear introduction for the importance and particularity of your research site.

Author response: This is a very good suggestion, and we thank the reviewer for making it. We now replaced the two sentences, talking instead about the importance of karst systems with regards to their carbon budget, and the fact that they can also emit an important amount of CO2 depending on whether or not there is a vegetation and the type of vegetation, hence underlining the importance of understanding the contribution of ecosystems to the carbon budget of karst areas.

We further added a related sentence in the introduction, so that everything written in the abstract has a source in the paper itself.

2. Introduction (Lines 47-55): the same problem with the point 1, process-based models and machine learning models are outside of your research. Thus, focus the introduction on the VI-based models and LUE models for GPP estimations.

Author response: Yes. We agree with this suggestion, that removes unnecessary developments, and keeps the introduction straight to the point. Accordingly, we removed the process-based models machine learning models from the introduction, and the introduction is more focused now on what was addressed in our study.

3. Results (Lines 257-259): Obviously, results of the best models selection are not showed in this part. Add some useful information in this part. In addition, considering a better model selection method because a simple comparison of statistical index (R2 and RMSE) is easy to make errors. For example, actually, it's hard to know which model is better for two models with similar R2, just like 0.82 and 0.8, or 0.73 and 0.72. (Fig. 3).

Author response: We thank the reviewer for this valuable comment. It is true that EVI and SAVI performed very similarly. We now computed the corrected AIC (or the AICc) which is ideal for model comparison for low sample sizes, while converging towards AIC for large samples. Therefore, the choice of AICc is the fact that it is suitable both for small and large sample sizes. AICc differences (ΔAICc) were computed between each candidate model and the model with the lowest AICc to support the other accuracy metrics in choosing the best model. According to Burnham and Anderson (2002) [reference included in the revised manuscript], a model can be said to be better than the other when the AICc difference is higher than 2. In this regard, while highlighting EVI as the best proxy for GPP for having the highest r² and the lowest rmse and AICc, we also highlighted the fact that based on ΔAICc < 2, SAVI performed similarly as EVI. For NEE however, the difference between EVI and SAVI was significant. More can be read about this in the revised manuscript.

Minor points:

1. Methods: add some information for your decision of footprint for EC-observations.

Author response: At first, we did not include any information about footprint analyses in our paper because the coauthor performed the analyses in a previous study that we referred to in our paper. However, we believe your suggestion will make the section of the study area easily understandable without necessarily going to the other paper. Therefore, we have now added a concise information about the footprint, especially the method used and the estimated average distance from where 90% of the fluxes were monitored by the tower.

2. Lines 95-97: “could be found in [24]”, it is a wrong way for references marking. You can rewrite it as “could be found in a previous study [24]”. So many similar references errors throughout the text, please check and revise.

Author response: Thank you for pointing out these errors in our referencing style. We corrected all the citations accordingly.

---

## [Decision Letter · Decision Letter 1]

9 Jul 2020

PONE-D-20-11727R1

Empirical vs. light-use efficiency modelling for estimating carbon fluxes in a mid-succession ecosystem developed on abandoned karst grassland

PLOS ONE

Dear Koffi Dodji Noumonvi,

Thank you for submitting your manuscript to PLOS ONE. After careful consideration, we feel that it has merit but does not fully meet PLOS ONE’s publication criteria as it currently stands. Therefore, we invite you to submit a revised version of the manuscript that addresses the points raised during the review process.

As you can see from the reviewers' comments at the end of this letter, the reviewers pointed out many low-graded errors in the manuscript. This makes me stronghly recommend you have a thorough polishing on the manuscript format and language. In addition to the major points raised by the second reviewer, the way how you deal with those low-graded errors will be the crucial critiria for me to make a final decision.

We look forward to receiving your revised manuscript.

Kind regards,

Wang Li

Academic Editor

PLOS ONE

Reviewers' comments:

Reviewer's Responses to Questions

**Comments to the Author**

1. If the authors have adequately addressed your comments raised in a previous round of review and you feel that this manuscript is now acceptable for publication, you may indicate that here to bypass the “Comments to the Author” section, enter your conflict of interest statement in the “Confidential to Editor” section, and submit your "Accept" recommendation.

Reviewer #1: All comments have been addressed

Reviewer #2: (No Response)

2. Is the manuscript technically sound, and do the data support the conclusions?

Reviewer #1: Yes

Reviewer #2: Yes

3. Has the statistical analysis been performed appropriately and rigorously? 

Reviewer #1: Yes

Reviewer #2: Yes

4. Have the authors made all data underlying the findings in their manuscript fully available?

Reviewer #1: Yes

Reviewer #2: Yes

5. Is the manuscript presented in an intelligible fashion and written in standard English?

Reviewer #1: Yes

Reviewer #2: Yes

6. Review Comments to the Author

Reviewer #1: The author has solved almost all the problems I have raised. The manuscript in the present form has been greatly improved. There are two minor problems:

Lines 20-21: This sentence ‘The CO2 released can be absorbed again for photosynthesis if there is a good vegetation cover on the site’ seems unnecessary.

Lines 292-296: These two paragraphs still have only one sentence, and they can be combined with the first paragraph (Lines 283-291).

Reviewer #2: General comments:

The authors have addressed some of my concerns. However, some flaws need to be improved.

Major points:

1. Some paragraphs are not well organized. Some examples:

(1) in the section of “Study site” (Lines 71-92), I think, there is no need to divide it into three segments.

(2) Line 112: it belongs to your results from data acquisition, thus should not appear in the methods section.

(3) Data analysis (Lines 187-210): can be expressed in only one paragraph, no need for three.

(4) Best models selection (Lines 283-286): it belongs to “methods”, not “Results”.

(5) Limitations of the study (Lines 407-420): express it in only one paragraph.

………………

Please check other similar questions.

2. Empirical models: in this study, why a simple quadratic model was selected to estimate GPP and NEE in this site, not others, just like simple linear regression, or exponential regression and so on? After multiple model comparisons? Or based on previous studies? Please specify it in the text.

Minor points:

1. References: so many errors in references marking, e.g., line 88, line 94, line 105, line 284…., Be careful, and not to make such a low-grade mistake again.

2. Title of the section (or sub-section): Don't be too simple to be accurately express your idea. e.g., Models (Line 130, GPP or NEE estimation models?), Empirical model (Line 212, Empirical model performance (in GPP or NEE estimation)?), fAPAR (Line 231, fAPAR estimation?)…….

7. PLOS authors have the option to publish the peer review history of their article (what does this mean?). If published, this will include your full peer review and any attached files.

Reviewer #1: No

Reviewer #2: No

---

## [Author Response · Author response to Decision Letter 1]

21 Jul 2020

We would like to thank the reviewers for their valuable comments, that has leveled our paper up in terms of consistency and coherence. Please find below a point by point response to all issues raised during the second round of review. We successfully addressed each of the comments in the manuscript, and changes can be noticed easily in the revised manuscript labelled “Revised Manuscript with Track Changes”.

Comments of the Reviewer 1

The author has solved almost all the problems I have raised. The manuscript in the present form has been greatly improved. There are two minor problems:

1. Lines 20-21: This sentence ‘The CO2 released can be absorbed again for photosynthesis if there is a good vegetation cover on the site’ seems unnecessary.

Author response: We thank the reviewer and have removed this unnecessary sentence that makes the abstract more concise.

2. Lines 292-296: These two paragraphs still have only one sentence, and they can be combined with the first paragraph (Lines 283-291).

Author response: This is absolutely a good suggestion, and we have joined these two paragraphs to the bigger previous one instead of having one sentence paragraphs which is not a good practice. We are very grateful to the reviewer for these valuable comments and suggestions

Comments of the Reviewer 2

The authors have addressed some of my concerns. However, some flaws need to be improved.

Major points:

1. Some paragraphs are not well organized. Some examples:

(1) in the section of “Study site” (Lines 71-92), I think, there is no need to divide it into three segments.

(2) Line 112: it belongs to your results from data acquisition, thus should not appear in the methods section.

(3) Data analysis (Lines 187-210): can be expressed in only one paragraph, no need for three.

(4) Best models selection (Lines 283-286): it belongs to “methods”, not “Results”.

(5) Limitations of the study (Lines 407-420): express it in only one paragraph.

………………

Please check other similar questions.

Author response: We thank the reviewer for these very important suggestions. We have now put together any unnecessarily split paragraphs (1), (3), (5) and other similar issues that we noticed by reading again the entire manuscript. The reviewer 1 also made such suggestions that we have taken into account. We have moved the text on model selection from results to integrate it in methods where it belongs (4). We have also moved the Figure 2 of Eddy covariance and vegetation indices to the results section and added some description to the newly created section. Additionally, we had the paper read by one peer, and considered his useful suggestions on language and phrasings as well.

With these changes, the manuscript is now more consistent and coherent, and we thank the reviewer for his useful comments.

2. Empirical models: in this study, why a simple quadratic model was selected to estimate GPP and NEE in this site, not others, just like simple linear regression, or exponential regression and so on? After multiple model comparisons? Or based on previous studies? Please specify it in the text.

Author response: It is true that we forgot to mention the reason for choosing a quadratic model. In fact, it was based on multiple model comparisons guided by the relationship observed between GPP or NEE and vegetation indices. We have now added this information in the methods sections where we presented the empirical model.

Minor points:

1. References: so many errors in references marking, e.g., line 88, line 94, line 105, line 284…., Be careful, and not to make such a low-grade mistake again.

Author response: We thank the reviewer for pointing out these citation issues. We thought it was okay to write the author’s name when we are referring to their model for instance. However, on the suggestion of the reviewer, we have now removed all author names, and rephrased differently either by using the phrasing “as in a previous study”, or by using passive voice and adding the reference at the end of the sentence. We have checked all citations in the text, and they are now all styled consistently, and comply with the citation style of the PLOS One journal.

2. Title of the section (or sub-section): Don't be too simple to accurately express your idea. e.g., Models (Line 130, GPP or NEE estimation models?), Empirical model (Line 212, Empirical model performance (in GPP or NEE estimation)?), fAPAR (Line 231, fAPAR estimation?).

Author response: We acknowledge that some of our titles are too simple and not clear enough. In this regard, we included the suggestions of the reviewer and have checked all titles across the document to make them more explicit. Apart from what was explicitly suggested by the reviewer, we also changed for instance the title “VPM model” to “VPM model performance in estimating GPP” (Line 256 of manuscript with track changes); the title “LUEmax” to “Estimated LUEmax” (Line 284 of manuscript with track changes).

---

## [Editor Report · Decision Letter 2]

24 Jul 2020

Empirical vs. light-use efficiency modelling for estimating carbon fluxes in a mid-succession ecosystem developed on abandoned karst grassland

PONE-D-20-11727R2

Dear Koffi Dodji Noumonvi,

We’re pleased to inform you that your manuscript has been judged scientifically suitable for publication and will be formally accepted for publication once it meets all outstanding technical requirements.

Kind regards,

Wang Li

Academic Editor

PLOS ONE
---

## [Editor Report · Acceptance letter]

29 Jul 2020

PONE-D-20-11727R2 

Empirical vs. light-use efficiency modelling for estimating carbon fluxes in a mid-succession ecosystem developed on abandoned karst grassland 

Dear Dr. NOUMONVI:

I'm pleased to inform you that your manuscript has been deemed suitable for publication in PLOS ONE. Congratulations! Your manuscript is now with our production department. 

Kind regards, 

on behalf of

Dr. Wang Li 

Academic Editor

PLOS ONE